# VDUP1 Deficiency Promotes the Severity of DSS-Induced Colitis in Mice by Inducing Macrophage Infiltration

**DOI:** 10.3390/ijms241713584

**Published:** 2023-09-01

**Authors:** Ki Hwan Park, Hyunju Lee, Hyoung-Chin Kim, Inpyo Choi, Sang-Bae Han, Jong Soon Kang

**Affiliations:** 1Laboratory Animal Resource Center, Korea Research Institute of Bioscience and Biotechnology, 30 Yeongudanji, Cheongwon, Cheongju 28116, Republic of Korea; brightnessd@kribb.re.kr (K.H.P.); hyunju35@kribb.re.kr (H.L.); hckim@kribb.re.kr (H.-C.K.); 2Immunotherapy Research Center, Korea Research Institute of Bioscience and Biotechnology, 125 Gwahak-ro Yuseoung-gu, Daejeon 34141, Republic of Korea; ipchoi@kribb.re.kr; 3College of Pharmacy, Chungbuk National University, 194-21 Osongsaengmyung-1-ro, Heungdeok-gu, Cheongju 28116, Republic of Korea; shan@chungbuk.ac.kr

**Keywords:** VDUP1, inflammation, macrophage infiltration, NF-κB, ulcerative colitis

## Abstract

The loss of vitamin D_3_ upregulated protein 1 (VDUP1) has been implicated in the pathogenesis of various inflammation-related diseases. Notably, reduced expression of VDUP1 has been observed in clinical specimens of ulcerative colitis (UC). However, the role of VDUP1 deficiency in colitis remains unclear. In this study, we investigated the role of VDUP1 in dextran sulfate sodium (DSS)-induced experimental colitis in mice. VDUP1-deficient mice were more susceptible to DSS-induced colitis than their wild-type (WT) littermates after 2% DSS administration. VDUP1-deficient mice exhibited an increased disease activity index (DAI) and histological scores, as well as significant colonic goblet cell loss and an increase in apoptotic cells. These changes were accompanied by a significant decrease in MUC2 mRNA expression and a marked increase in proinflammatory cytokines and chemokines within damaged tissues. Furthermore, phosphorylated NF-κB p65 expression was significantly upregulated in damaged tissues in the context of VDUP1 deficiency. VDUP1 deficiency also led to significant infiltration of macrophages into the site of ulceration. An in vitro chemotaxis assay confirmed that VDUP1 deficiency enhanced bone marrow-derived macrophage (BMDM) chemotaxis induced by CCL2. Overall, this study highlights VDUP1 as a regulator of UC pathogenesis and a potential target for the future development of therapeutic strategies.

## 1. Introduction

Ulcerative colitis (UC) is a chronic inflammatory bowel disease (IBD) characterized by symptoms such as diarrhea, abdominal pain, and rectal bleeding [1]. The pathogenesis of UC involves various factors, including genetic predisposition, environmental influences, microbial interactions, and dysregulated inflammatory responses [2,3,4]. Dysfunctions in intestinal epithelial cell (IEC) turnover and the goblet cell mucus layer have been associated with the progression of UC [5,6].

Furthermore, immune cell infiltration, particularly that of macrophages, in the affected tissues of UC patients contributes to the excessive expression of proinflammatory cytokines and chemokines, which play a critical role in abnormal immune responses [7,8,9].

Previous reports suggested that the NF-κB pathway plays a crucial role in the pathogenesis of UC. NF-κB activation is implicated in the aberrant expression of inflammatory cytokines and monocyte-attracting chemokines [10,11,12]. Indeed, macrophages and epithelial cells isolated from the inflamed intestinal tissues of human UC specimens showed increased levels of NF-κB activation [13]. Therefore, numerous studies are being conducted to investigate genes and drugs that interact with NF-κB in the context of UC [11]. Despite extensive research, many aspects of UC onset and development are still not fully understood. Moreover, the infiltration of immune cells, including macrophages and neutrophils, which are largely influenced by the NF-κB signaling pathway, is closely associated with the onset of UC [13,14,15]. In fact, pharmacological interventions to inhibit the responsiveness of macrophages to CCL2 are being proposed as a therapeutic approach for UC [16].

Vitamin D_3_ upregulated protein 1 (VDUP1), which is also known as thioredoxin-interacting protein (TXNIP) or thioredoxin-binding protein (TBP-2), was initially identified as an upregulated gene in HL-60 cells treated with 1,25-dihydroxyvitamin D_3_ [17]. Several groups suggest that the upregulation of VDUP1 is involved in the initiation and progression of oxidative stress-related disorders such as diabetic diseases [18,19,20,21,22,23,24], cardiovascular disease [25], and neurological disorders [26,27,28,29,30]. Conversely, several studies have reported that VDUP1 protects against various diseases, including steatohepatitis [31], hepatocarcinogenesis [32], gastritis, and gastric carcinogenesis [33], in animal models. The role of VDUP1 as a promoter or suppressor of inflammation and inflammation-related disorders is controversial, as it manifests differently in various diseases, suggesting the existence of complex crosstalk specific to each disease. Consistent with this, extensive research is concurrently being conducted to investigate VDUP1 inhibition and activation in the treatment of diseases [34,35,36].

VDUP1 expression is decreased in colonic mucosa specimens of UC and colorectal cancer (CRC), suggesting its involvement in the pathogenesis of UC [37]. However, the precise role of VDUP1 in UC remains largely unknown. In this study, we investigated the role of VDUP1 in a dextran sulfate sodium (DSS)-induced experimental colitis model. We evaluated histopathological changes and the expression of inflammatory mediators. We also examined NF-kB activation and macrophage infiltration at the inflamed site. Our findings suggest a critical role of VDUP1 in the pathogenesis of UC.

## 2. Results

### 2.1. The Expression of VDUP1 Was Reduced in Experimental Colitis

To establish the correlation between clinical specimens and animal experiments, we investigated the colonic expression of VDUP1 in an experimental colitis model using WT mice. The mRNA (Figure 1B) and protein (Figure 1C) levels of VDUP1 in WT mice were lower in the inflamed tissues in the DSS-induced colitis model compared to the untreated control.

### 2.2. VDUP1 Deficiency Exacerbated the Severity of DSS-Induced Colitis

To further elucidate the involvement of VDUP1 in UC, we investigated the effects of VDUP1 loss in an acute colitis model induced by DSS, which mimics the clinical features of UC. WT and VDUP1-KO mice exhibited significant weight loss following DSS treatment (Figure 2A). On the other hand, the VDUP1-KO mice exhibited significantly more weight loss than the WT mice (Figure 2A). The body weight changes for WT + 2% DSS mice were as follows: 100.0 ± 0.0% (day 0), 100.1 ± 0.7% (day 1), 99.0 ± 0.8% (day 2), 100.0 ± 0.7% (day 3), 100.6 ± 0.6% (day 4), 99.9 ± 0.8% (day 5), 96.3 ± 0.8% (day 6), 90.3 ± 3.0% (day 7), 89.4 ± 1.8% (day 8), and 88.1 ± 2.1% (day 9). For VDUP1-KO + 2% DSS mice, the corresponding values were 100.0 ± 0.0% (day 0), 100.0 ± 0.7% (day 1), 100.6 ± 0.8% (day 2), 102.1 ± 0.6% (day 3), 102.2 ± 0.6% (day 4), 99.4 ± 1.3% (day 5), 92.2 ± 1.9% (day 6), 86.0 ± 2.0% (day 7), 81.0 ± 2.0% (day 8), and 77.5 ± 2.1% (day 9). No significant body weight changes were observed in the control group of WT and VDUP1-KO mice. Moreover, the VDUP1-KO mice exhibited more severe features of the disease, as measured by the disease activity index (DAI) score (Figure 2B), which is the sum of the body weight score (Appendix A), stool consistency score (Appendix A), and rectal bleeding score (Appendix A). The DAI scores for WT + 2% DSS were as follows: 0.0 ± 0.0 (day 0), 1.4 ± 0.2 (day 1), 1.6 ± 0.4 (day 2), 1.2 ± 0.3 (day 3), 2.0 ± 0.4 (day 4), 2.4 ± 0.3 (day 5), 3.6 ± 0.4 (day 6), 4.9 ± 0.4 (day 7), 5.2 ± 0.5 (day 8), and 4.9 ± 0.4 (day 9). For VDUP1-KO + 2% DSS mice, the corresponding values were 0.0 ± 0.0 (day 0), 1.7 ± 0.2 (day 1), 2.3 ± 0.3 (day 2), 2.4 ± 0.4 (day 3), 3.0 ± 0.4 (day 4), 3.3 ± 0.3 (day 5), 6.1 ± 0.8 (day 6), 6.4 ± 0.5 (day 7), 8.0 ± 0.5 (day 8), 6.8 ± 0.4 (day 9). No significant disease activities were observed in the control group of WT and VDUP1-KO mice. Furthermore, the histological score (Figure 2D), which is the sum of the inflammation score (Appendix A), epithelial defects score (Appendix A), and crypt atrophy score (Appendix A), was higher in the VDUP1-KO mice than in WT mice.

### 2.3. VDUP1 Deficiency Accelerated Colonic Tissue Damage in Experimental Colitis

We then assessed whether VDUP1 deficiency promoted crypt damage in DSS-induced colitis. We conducted a terminal deoxynucleotidyl transferase dUTP nick end labeling (TUNEL) assay, and the results revealed that VDUP1 deficiency significantly increased apoptosis in the damaged tissues compared to that in WT mice (Figure 3A,B). Goblet cells play a crucial role in maintaining colonic homeostasis by secreting gel-forming mucins and forming the structural framework of the mucus layer in the gut [38]. Therefore, we evaluated whether VDUP1 deficiency affected goblet cell defects. Alcian blue/periodic acid-Schiff (AB-PAS) staining showed a significant decrease in the number of goblet cells in the colons of VDUP1-KO mice (Figure 3C,D). Consistent with goblet cell loss, the mRNA level of MUC2 in the colons was significantly reduced in VDUP1-KO mice (Figure 3E). Treatment of WT and VDUP1-KO mice with 2% DSS resulted in a 23% and 88% inhibition of MUC2 mRNA expression, respectively, to compare untreated control.

### 2.4. VDUP1 Deficiency Induced Inflammatory Cytokine Expression in Experimental Colitis

The severity of inflammation in experimental colitis can be assessed by the expression of inflammatory mediators. In this study, the mRNA expression of inflammatory mediators, including interleukin-6 (IL-6) (Figure 4A), tumor necrosis factor (TNF)-α (Figure 4B), interleukin-1β(IL-1β) (Figure 4C), and COX-2 (Figure 4D) was significantly increased in the colons of VDUP1-KO mice compared to WT mice. Treatment of VDUP1-KO mice with 2% DSS resulted in a 732%, 140%, 199%, and 476% increase in mRNA expression of IL-6, TNF-α, IL-1β, and COX-2, respectively, compared to those in WT mice. No significant difference in mRNA expression of IL-6, TNF- α, IL-1β, and COX-2 was observed between the control group of WT and VDUP1-KO mice.

### 2.5. VDUP1 Deficiency Activated NF-κB p65 in Experimental Colitis

Immunohistochemical staining revealed the expression of phosphorylated p65 (p-p65) in the untreated control group, villi, and ulcerated areas in DSS-treated WT and VDUP1-KO mice (Figure 5A,B). In the control groups of VDUP1-KO mice and WT mice, there was minimal expression of p-p65 in the villi. However, after DSS administration, a significant increase in p-p65 expression was observed specifically in the villi of VDUP1-KO mice compared to mice in the control group that did not receive DSS. Moreover, in the ulcerated areas, WT mice and VDUP1-KO mice showed a significant increase in p-p65 expression compared to that in the control. Notably, VDUP1-KO mice exhibited a marked increase in p-p65 compared to WT mice in these ulcerated areas.

### 2.6. VDUP1 Deficiency Promoted Macrophage Chemotaxis to the Site of Inflammation

During the pathogenesis of IBD, pathogens cross the damaged intestinal epithelial barrier and activate macrophages, which produce proinflammatory cytokines such as IL-1β, IL-6, and TNF-α. These cytokines, in turn, act directly or indirectly on intestinal epithelial cells, causing injury or necrosis. Histological analysis revealed a significant increase in F4/80-positive macrophage infiltration in the ulceration areas in VDUP1-KO mice compared to WT mice (Figure 6A,B), which was accompanied by markedly increased mRNA expression of F4/80 (Figure 6C). Treatment of VDUP1-KO mice with 2% DSS resulted in a 153% increase in mRNA expression of F4/80 compared to that in WT mice. No significant difference in mRNA expression of F4/80 was observed between the control group of WT and VDUP1-KO mice. Furthermore, VDUP1-KO mice exhibited increased mRNA expression of macrophage-attractive chemokines, including C-C motif chemokine ligand 2 (CCL2/MCP-1) (Figure 6D), C-C motif chemokine ligand 3 (CCL3/MIP1A) (Figure 6E), and keratinocyte chemoattractant (KC/CXCL1) (Figure 6F) in mice with DSS-induced colitis compared to WT mice. Treatment of VDUP1-KO mice with 2% DSS resulted in a 218%, 920%, and 664% increase in mRNA expression of CCL2, CCL3, and KC, respectively, compared to those in WT mice. No significant difference in mRNA expression of CCL2, CCL3, and KC was observed between the control group of WT and VDUP1-KO mice. To further investigate the involvement of VDUP1 in macrophage chemotaxis, we performed a chemotaxis assay using bone marrow-derived macrophages (BMDMs) stimulated with recombinant mouse CCL2 (rmCCL2) as a chemoattractant. Flow cytometry revealed that there was no difference in the F4/80-positive macrophage differentiation rate between WT and VDUP1-KO mice (Appendix A). BMDMs derived from VDUP1-KO mice exhibited significantly increased migration in response to rmCCL2 (Figure 6G,H). Moreover, the mRNA expression of CCL2 was markedly upregulated in lipopolysaccharide (LPS)-stimulated BMDMs from VDUP1-KO mice compared to BMDMs from WT littermates (Figure 6I). Treatment of BMDMs from VDUP1-KO mice caused a 166% increase in mRNA expression of CCL2 compared to that in BMDMs from WT mice. No significant difference in mRNA expression of CCL2 was observed between the control group of WT and VDUP1-KO mice.

## 3. Discussion

Previous studies have demonstrated the significant involvement of VDUP1 in various disease pathogeneses, including diabetes [22], cardiovascular diseases [39], Alzheimer’s disease [28], endotoxic shock [40], nonalcoholic steatohepatitis (NASH) [31], gastric carcinogenesis [33], and bladder carcinogenesis [41] using animal disease models. However, the role of VDUP1 in distinct diseases is not universally consistent. Several studies have indicated that the VDUP1/NOD-like receptor family protein 3 (NLRP3) pathway contributes to the development of diabetes [22], cardiovascular diseases [39], and Alzheimer’s disease [28]. Zhou et al. reported that VDUP1 is released from oxidized thioredoxin (TRX) and binds to the NLRP3 inflammasome, activating the release of IL-1β and IL-18, thereby triggering an inflammatory response [42]. Conversely, several groups have proposed that VDUP1 inhibits the pathogenesis of endotoxic shock [40], NASH [31], gastric carcinogenesis [33], and bladder carcinogenesis [41] by suppressing the inducible nitric oxide synthase (iNOS), mechanistic target of rapamycin kinase (MTOR), NF-κB, and extracellular signal-regulated kinase (ERK), respectively. Thus, VDUP1 is acknowledged as an ambivalent gene, and maintaining a balance in the expression and function of VDUP1 is crucial [43].

Our findings suggested an inverse correlation between VDUP1 and colitis. A previous study demonstrated decreased expression of VDUP1 in human UC specimens [37]. The chemically induced colitis mouse model using DSS is widely used due to its ability to reflect pathophysiological features and histological changes observed in humans [44,45,46]. Therefore, we used this experimental colitis model to demonstrate the same phenomenon observed in humans, providing a translational aspect. In our study, VDUP1-deficient mice exhibited more severe colitis than WT mice following DSS administration. Furthermore, VDUP1-KO mice exhibited a significantly increased inflammatory response, characterized by macrophage infiltration, during DSS-induced colitis.

It is well known that relatively thin and discontinuous mucus layer, the depletion of goblet cells, and reduced MUC2 expression are associated with the development of UC [47,48,49]. MUC2-deficient mice spontaneously develop colitis, highlighting the crucial role of MUC2 in protecting against colitis [50]. In this study, we observed significant disruption of goblet cells and a marked increase in apoptotic cells in the inflamed tissues of VDUP1-KO mice compared to WT mice with DSS-induced colitis. Consistent with these findings, the mRNA level of MUC2 was significantly reduced in VDUP1-KO mice after DSS administration compared to WT mice. Moreover, DSS-treated VDUP1-KO mice exhibited severe colitis, as evidenced by several clinicopathological indicators, including more pronounced weight loss, diarrhea, and rectal bleeding, as well as histological indicators, such as inflammation, epithelial defects, and crypt atrophy. These findings suggest that the loss of VDUP1 induces physiological damage during the development of DSS-induced colitis.

Colonic barrier dysfunction mediates a breach in the damaged intestinal epithelial cell barrier by pathogens, stimulating lamina propria cells to produce proinflammatory cytokines such as TNF-α, IL-6, and IL-1β [14]. These cytokines can directly act on intestinal epithelial cells, leading to cell injury or necrosis, thereby promoting the occurrence and progression of UC. Increasing evidence suggests that inhibiting TNF-α, IL-6, and IL-1β ameliorates the development of UC [51,52,53,54]. In this study, we observed a significant increase in the mRNA levels of proinflammatory cytokines, including TNFα, IL-6, and IL-1β, in the colon tissues of VDUP1-KO mice following DSS treatment compared to WT mice. These findings indicate that VDUP1 deficiency plays a critical role in creating an inflammatory environment and promotes the expression of inflammatory cytokines in DSS-induced colitis.

It has been reported that inhibition of the NF-κB signaling pathway, which controls the expression of proinflammatory cytokines and chemokines, ameliorates the development of UC in clinical and preclinical studies [15,55,56]. Moreover, previous reports have demonstrated that VDUP1 inhibits hepatocarcinogenesis by suppressing TNF-α-induced NF-κB activation [32]. Additionally, Kim et al. demonstrated that VDUP1 could reduce the migration of ovarian cancer cells by inhibiting NF-κB activation [57]. However, whether VDUP1 suppresses the NF-κB signaling pathway in UC remains unclear. In this study, immunohistochemical analysis revealed that the loss of VDUP1 significantly increased the nuclear expression of p65 phosphorylated at serine 536 in the colonic tissues of the experimental colitis model, compared to WT mice. These findings indicate that VDUP1 deficiency plays a crucial role in the activation of NF-κB and downstream inflammatory mediators in experimental colitis.

A previous report suggested that blocking the chemokine CCL2, which is known to impede macrophage infiltration, can prevent the development of colitis and colitis-associated carcinogenesis in mice [58]. In this study, we demonstrated that the loss of VDUP1 enhanced macrophage infiltration in DSS-induced colitis, which was accompanied by a significant increase in the mRNA expression of macrophage-attracting chemokines, including CCL2, CCL3, and KC in inflamed colon tissues. Moreover, we observed a significant increase in the in vitro migration of BMDMs obtained from VDUP1-KO mice in response to stimulation with rmCCL2 compared to BMDMs from WT mice. Interestingly, the mRNA expression of CCL2 was significantly increased in LPS-stimulated BMDMs obtained from VDUP1-KO mice compared to WT mice. These findings demonstrate that VDUP1 deficiency increases the expression of chemokines in the inflammatory environment and explains the enhanced chemotactic responsiveness of macrophages toward CCL2.

However, several limitations of the current study should be noted. First, our investigation exclusively focused on analyzing the role of VDUP1 in an acute UC. However, in the context of translational research, investigating chronic UC and UC-mediated carcinogenesis could provide a more comprehensive understanding of specific gene functions [59,60,61]. Second, we used pan-VDUP1-KO mice in the DSS-induced colitis model. To identify the role of particular genes in the pathogenesis of UC, organ- or cell-type-specific knockout animals are commonly employed [62,63,64]. Third, we did not investigate a specific agonist for VDUP1. Target-specific agonists could demonstrate the therapeutic potential of specific genes in UC pathogenesis [65,66,67]. However, these issues will be addressed in our subsequent study.

When taken together, the deficiency of VDUP1 in the DSS-induced colitis model was found to induce excessive tissue damage in the colon, accompanied by significant NF-κB activation, inflammation, and macrophage chemotaxis. This study elucidates the pathogenesis of the colitis associated with VDUP1, and future work will focus on exploring the potential role of VDUP1 in treating UC.

## 4. Materials and Methods

### 4.1. Reagents and Animals

Unless otherwise noted, all reagents were purchased from Sigma-Aldrich (St. Louis, MO, USA). VDUP1-KO mice were generated as described previously [68]. These mice were maintained with the C57BL/6 strain and backcrossed with C57BL/6 for more than 10 generations. The mice used in this study were free of antibodies for 14 murine viruses and negative for pathogenic bacteria and parasites. The in vivo animal study was approved by the Institutional Animal Care and Use Committee of the Korea Research Institute of Bioscience and Biotechnology (approval number: KRIBB-AEC-13165).

### 4.2. DSS-Induced Colitis

For DSS-induced colitis, 7-week-old mice were administered 2% DSS (33–55 kDa, TdB consultancy, Uppsala, Sweden) in drinking water for 5 days, followed by 4 days of normal water. The body weight, stool consistency, and presence of blood in stool were recorded daily. Parameters for the disease activity index (DAI), including weight loss, stool consistency, and rectal bleeding, were scored according to the indicated criteria (Appendix A).

### 4.3. RNA Isolation and Quantification of mRNA Expression

Total RNA was extracted from colon tissues and BMDMs by using TRIzol reagents (Ambion, Austin, TX, USA) according to the manufacturer’s protocols. Briefly, tissue and cells were collected and homogenized in TRIzol reagent. Chloroform was added to the homogenate and mixed thoroughly. The mixture was centrifuged to separate the phases. The aqueous phase containing RNA was transferred to a new tube. Isopropanol was added to precipitate RNA, followed by centrifugation. The RNA pellet was washed with ethanol, air-dried, and dissolved in RNase-free water. The quality and quantity of RNA were determined by measuring the absorbance at 260 and 280 nm using infiniteM200 (TECAN, Switzerland). Single-strand cDNA was generated from 1 μg of total RNA by reverse transcription using AccuPower RT PreMix (Bioneer, Daejeon, Korea) according to the manufacturer’s protocols (42 °C for 60 min and 95 °C for 5 min). The resulting cDNA was amplified by qRT-PCR with Power SYBR Green PCR Master Mix (Invitrogen, Carlsbad, CA, USA) in a thermal cycler. (For qRT-PCR, samples were amplified by 45 cycles of denaturation (95 °C for 15 s) and amplification (60 °C for 1 min) using ABI 7500 Fast Real-Time PCR System (Applied Biosciences, Foster City, CA, USA).) The gene expression levels relative to the control gene (β-actin) were calculated by the 2^−△△Ct^ method. All primer sequences are listed in Appendix A.

### 4.4. Colon Histology, Immunohistochemistry, Immunofluorescence Analysis, and TUNEL Assay

Colons were freshly collected, washed with 1X PBS, longitudinally cut, and positioned as a Swiss roll in 10% buffered formalin. The fixed samples were embedded in paraffin, sectioned, and stained with hematoxylin and eosin (H&E). H&E-stained sections of colon tissue from WT and VDUP1-KO mice were scored in a blinded manner. Histological scores were evaluated according to a previously published system (Appendix A) [69]. Immunohistochemical analysis was performed using an Avidin–Biotin Complex Staining Kit (Vector Laboratories, Burlingame, CA, USA). After being labeled with specific antibodies, the sections were developed with a DAB Substrate Kit (Vector Laboratories, Burlingame, CA, USA). For immunofluorescence analysis, tissue sections were deparaffinized, rehydrated, and permeabilized with PBS containing 0.2% Triton X-100 for 10 min. The sections were washed in 1X PBS three times for 5 min, blocked for 1 h with normal 10% serum from the same species as the secondary antibody, washed in PBS containing 0.05% Tween 20 (PBST), incubated with primary antibodies overnight at 4 °C, incubated with secondary antibodies conjugated with fluorescein for 1 h at room temperature in the dark, washed with PBST in the dark, counterstained with 0.1 μg/mL 4′, 6′-diamidino-2-phenylindole (DAPI), and rinsed with PBS. The antibodies used in this study are listed in Appendix A. Apoptotic cells were examined using a TUNEL Apoptosis Detection Kit (Millipore Corporation, Billerica, MA, USA).

### 4.5. AB-PAS Staining

For AB-PAS staining, tissue sections were deparaffinized, rehydrated, stained with alcian blue solution for 30 min, washed with running tap water, rinsed with distilled water (DW), treated with periodic acid for 5 min, washed with DW, stained with Schiff’s reagent for 10 min, washed with running tap water, counterstained with hematoxylin for 1 min, dehydrated with an ethanol gradient and xylene, and mounted with Canada balsam. Deep blue and clear blue indicate goblet cells and acidic mucin, respectively.

### 4.6. Primary Cell Culture and Chemotaxis Assay

To prepare BMDMs, BM cells were isolated from 6- to 8-week-old male C57BL6 mice and VDUP1-KO mice and maintained as previously described manner [70]. For qRT-PCR, BMDMs were incubated with 1 µg/mL LPS for 8 h. For the chemotaxis assay, 10 ng/mL rmCCL2 (R&D Systems, Minneapolis, MI, USA) was placed in the bottom chamber of a 24-well plate. BMDMs were seeded at a density of 1 × 10^5^ cells/well in the upper chamber (5 µm, Corning Incorporated, NY, USA). Each assay condition was performed in triplicate. After 18 h, the migrated cells were fixed with 4% paraformaldehyde, and the membranes were stained with 0.5% crystal violet (*w*/*v*) dissolved in 20% methanol (*v*/*v*) in PBS for 20 min. Five random views were photographed in each well at 100× magnification with a microscope. The number of migrated cells in five random fields was counted, and data from each membrane were averaged.

### 4.7. Statistical Analysis

The results are expressed as mean ± SEM. Two-way and three-way ANOVA followed by Tukey’s multiple comparisons test or unpaired *t*-test was used for statistical analysis using GraphPad Prism 10.0.2 (GraphPad Software, La Jolla, CA, USA). The criteria for statistical significance were set at ^§^
*p* < 0.05; ^§§^
*p* < 0.01; ^§§§^
*p* < 0.001; ^§§§§^
*p* < 0.0001; * *p* < 0.05; ** *p* < 0.01; *** *p* < 0.001; **** *p* < 0.0001; and ns., not significant.

## Figures and Tables

**Figure 1 ijms-24-13584-f001:**
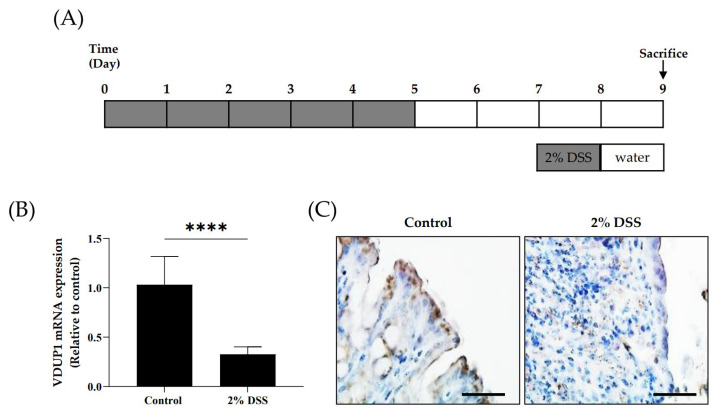
VDUP1 protein and mRNA expression were reduced in DSS-induced experimental colitis. (**A**) Colitis was induced in WT mice by the administration of 2% DSS in the drinking water for 5 days, followed by regular drinking water for an additional 4 days. (**B**) VDUP1 mRNA levels in the colon tissues of WT mice treated with DSS were detected by using qRT-PCR (*n* = 10). (**C**) Immunohistochemical staining of VDUP1 in the colon tissues of WT mice (40X). The data are expressed as the mean ± SEM. Scale bars, for B 50 µm. **** *p* < 0.0001.

**Figure 2 ijms-24-13584-f002:**
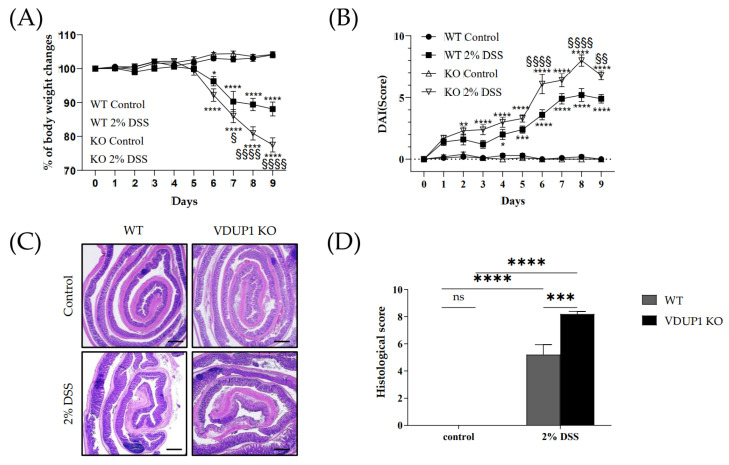
VDUP1 deficiency enhanced the severity of dextran sulfate sodium (DSS)-induced colitis. Colitis was induced in WT and VDUP1-KO mice by the administration of 2% DSS in the drinking water for 5 days, followed by regular drinking water for an additional 4 days (*n* = 9). The mice were evaluated daily for (**A**) percent body weight changes (**B**) and disease activity index (DAI) scores using the indicated criteria (Appendix A) (*n* = 9). Colons were collected on day 9 after DSS administration. (**C**) Representative histological colon sections stained with hematoxylin and eosin (H&E) on day 9 after colitis induction are shown (4 X) (*n* = 5). (**D**) Histological scores were blindly scored according to the indicated criteria (Appendix A) (*n* = 5). Scale bars, for E 500 µm. The data are expressed as the mean ± SEM. For panel (**A**) and (**B**), ^§^
*p* < 0.05, ^§§^
*p* < 0.01, ^§§§§^
*p* < 0.0001 for WT 2% DSS vs. VDUP1-KO 2% DSS; * *p* < 0.05, ** *p* < 0.01, *** *p* < 0.001, **** *p* < 0.0001 for control vs. 2% DSS; for panel (**D**), *** *p* < 0.001, **** *p* < 0.0001; ns, not significant.

**Figure 3 ijms-24-13584-f003:**
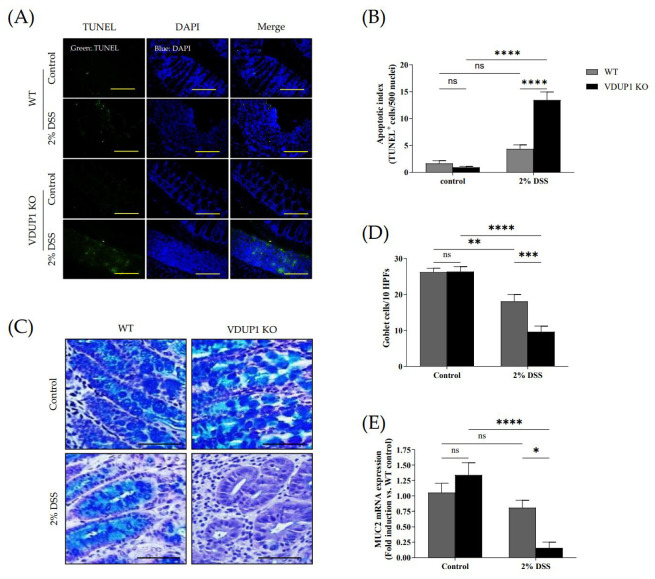
VDUP1 deficiency promotes apoptosis and goblet cell defects in the colon in DSS-induced colitis. Colons were collected on day 9 after DSS administration. Representative photomicrographs of (**A**) TUNEL assays (20 X) (*n* = 4) and (**C**) AB/PAS staining (400 X) (*n* = 5) are shown. Quantification of (**B**) apoptotic cells (TUNEL-positive cells per 500 nuclei) (**D**) and goblet cells (AB/PAS-positive cells per ten high power fields per mouse) are presented. (**E**) MUC2 mRNA levels in colon tissues were determined using qRT-PCR (*n* = 4). Scale bars for (**A**), 100 µm; for (**C**), 200 µm. The data are expressed as the mean ± SEM. * *p* < 0.05; ** *p* < 0.01; *** *p* < 0.001; **** *p* < 0.0001; ns, not significant.

**Figure 4 ijms-24-13584-f004:**
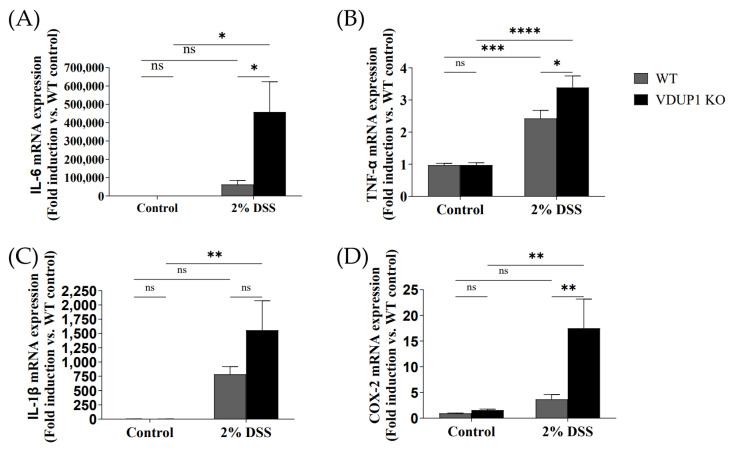
VDUP1 deficiency induces downstream proinflammatory mediators of the NF-κB signaling pathway in DSS-induced colitis. Colons were collected on day 9 after DSS administration. The mRNA expression of (**A**) IL-6, (**B**) TNF-α, (**C**) IL-1β, and (**D**) COX-2 in the colons was analyzed using qRT-PCR (*n* = 4). All of the data are presented as the mean ± SEM. * *p* < 0.05; ** *p* < 0.01; *** *p* < 0.001; **** *p* < 0.0001; ns, not significant.

**Figure 5 ijms-24-13584-f005:**
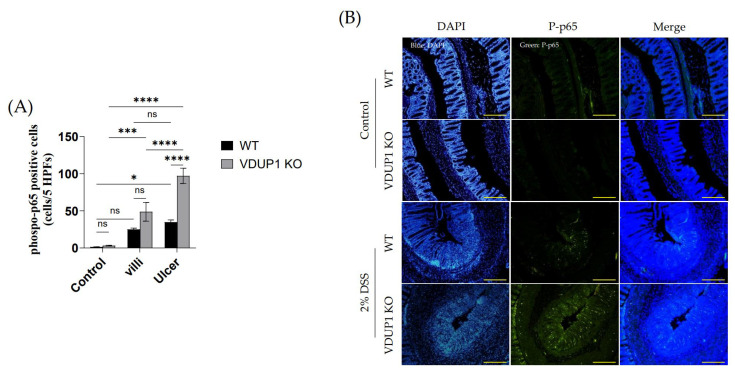
VDUP1-deficiency increases the expression of phosphorylated p65 in DSS-induced colitis. Colons were collected on day 9 after DSS administration. (**B**) Immunofluorescence staining was used to determine the expression of p-p65 (20 X) (*n* = 4). (**A**) The number of p-p65 positive cells was quantified (p-p65 positive cells per five high-power fields per mouse) (*n* = 4). Scale bars for (**B**) 100 µm. The data are expressed as the mean ± SEM. * *p* < 0.05; *** *p* < 0.001; **** *p* < 0.0001; ns, not significant.

**Figure 6 ijms-24-13584-f006:**
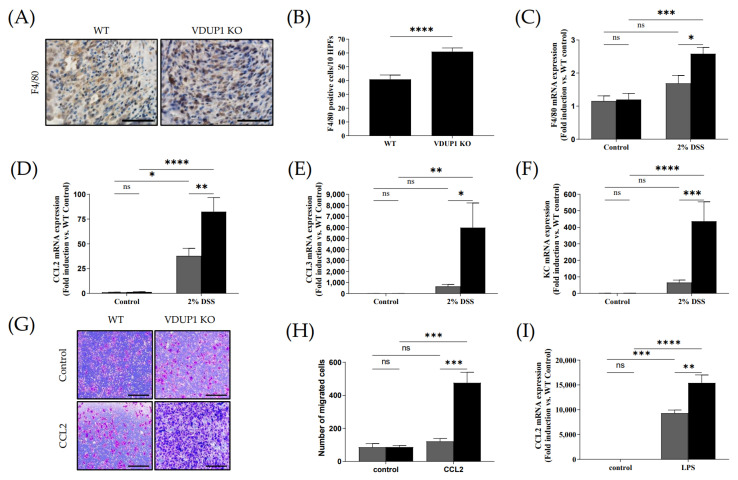
VDUP1 deficiency promotes macrophage migration to the site of inflammation in DSS-induced colitis. Colons were collected on day 9 after DSS administration. (**A**) Immunohistochemical staining (40 X) was performed to assess F4/80 expression (*n* = 4). (**B**) The number of F4/80-positive cells was quantified (ten high power fields per mouse) (*n* = 4). The mRNA expression of (**C**) F4/80, (**D**) CCL2, (**E**) CCL3, and (**F**) KC was determined using qRT-PCR (*n* = 4). Ex vivo differentiation of bone marrow cells from WT and VDUP1-KO mice is shown in Appendix A. Representative photomicrographs show migrated cells induced by (**G**) rmCCL2 (10X) (*n* = 3). The number of migrated cells in response to (**H**) rmCCL2 was counted (*n* = 3). (**I**) The mRNA expression of CCL2 in LPS-stimulated BMDMs was determined by qRT-PCR (*n* = 3). Scale bars for (**A**), 50 µm; for (**G**), 100 µm. The data are expressed as the mean ± SEM. * *p* < 0.05; ** *p* < 0.01; *** *p* < 0.001; **** *p* < 0.0001; ns, not significant.

## Data Availability

Not applicable.

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
