# Peer review of "VDUP1 Deficiency Promotes the Severity of DSS-Induced Colitis in Mice by Inducing Macrophage Infiltration"

_ijms, 2023, doi:10.3390/ijms241713584_

Round 1

Reviewer 1 Report

Title: Role of Vitamin D3 Upregulated Protein 1 (VDUP1) in Experimental Colitis: Findings, Limitations, and Therapeutic Implications

Introduction: The article investigates the role of Vitamin D3 Upregulated Protein 1 (VDUP1) in ulcerative colitis (UC), a chronic inflammatory bowel disease. The study highlights the involvement of VDUP1 in UC onset and development, particularly in macrophage activation and inflammatory responses. The authors aim to clarify the role of VDUP1 in UC pathogenesis using a dextran sulfate sodium (DSS)-induced experimental colitis model.

Experimental Design and Methodology: The study utilizes a DSS-induced colitis model in mice to investigate the role of VDUP1 in UC pathogenesis. VDUP1-KO mice and wild-type (WT) mice were used, and various assessments were performed, including measuring weight loss, disease activity index (DAI), histological changes, apoptotic cell levels, goblet cell defects, inflammatory mediator expression, NF-κB activation, and macrophage chemotaxis. The statistical analysis involved two-way ANOVA with Tukey's multiple comparisons test or unpaired t-test.

Results and Findings:

1)      VDUP1 expression was reduced in experimental colitis, suggesting an inverse correlation with the disease.

2)      VDUP1 deficiency exacerbated the severity of DSS-induced colitis, resulting in more weight loss, higher DAI scores, and increased histological damage compared to WT mice.

3)      VDUP1 deficiency accelerated colonic tissue damage in experimental colitis, leading to increased apoptosis and reduced goblet cells.

4)      VDUP1 deficiency induced the expression of inflammatory cytokines in experimental colitis, contributing to increased inflammation.

5)      VDUP1 deficiency activated NF-κB p65 in experimental colitis, implicating VDUP1 in regulating the NF-κB signaling pathway.

6)      VDUP1 deficiency promoted macrophage chemotaxis to the site of inflammation, linking VDUP1 to macrophage infiltration and inflammation.

 Limitations and Experimental Flaws:

           1)    Small Sample Size and Lack of Replication:

The study does not mention the sample size used in some experiments, which raises concerns about statistical power. Additionally, in some experiments, the number of replicates used is either low or not mentioned at all, which is making it difficult to assess the statistical robustness of the findings.

 2)    Lack of Detailed Experimental Protocols:

The study lacks comprehensive descriptions of the experimental protocols, including details on sample preparation, RNA isolation, and qRT-PCR conditions. Insufficient information may hinder reproducibility and interpretation of results. In the gene expression analysis by QRT-PCR, B-actin was selected as the housekeeping (reference) gene for relative expression quantification. However, Nour Eissa et al 2017 recently reported that B-actin and Gapdh is less stable and highly variable housekeeping (reference) gene in DSS-induced colitis C57BL/6J mice, and TATA-box-binding protein (Tbp) and eukaryotic translation elongation factor 2 (Eef2)are most suitable as housekeeping (reference) genes. https://www.nature.com/articles/srep42427. The studies show that intestinal inflammation significantly affects the stability of mucosal Gapdh, Actb and β2m expression, which displays high variability in healthy individuals and/or between the non-inflamed and inflamed mucosa and that normalization of colonic TNF-α and IL-1β mRNA levels significantly dependents on the stability of reference genes. Therefore, the author should re-qualify the gene expression with a more appropriate reference gene or genes.

 3)      Lack of Control Groups:

The study lacks control groups for some analyses, which could limit the ability to draw robust conclusions. Including appropriate control, groups would strengthen the experimental design.

 4)      Insufficient Method Validation:

The study does not provide validation of key methods used, such as the chemotaxis assay and TUNEL assay. Validation experiments would enhance the reliability of the results.

 5)      Limited Mechanistic Insights:

The study does not provide detailed mechanistic insights into how VDUP1 deficiency leads to increased inflammation, macrophage infiltration, and NF-κB activation. A more in-depth analysis of the underlying molecular mechanisms would strengthen the study's conclusions.

 6)      Limited Investigation of VDUP1 Activation:

While the study focuses on VDUP1 deficiency, it fails to explore the role of VDUP1 activation. A comprehensive understanding of both VDUP1 deficiency and activation could provide a more holistic view of VDUP1's involvement in UC.

7)      Insufficient Discussion of Conflicting Findings:

The study acknowledges the controversial role of VDUP1 in inflammation-related disorders but does not thoroughly discuss the possible reasons for these conflicting findings in different diseases.

 8)      Incomplete NF-κB Activation Analysis:

The study reports increased NF-κB p65 activation in VDUP1-KO mice, but a more in-depth analysis of the downstream targets and functional consequences of NF-κB activation would provide a clearer understanding of its role in UC pathogenesis.

9)      Lack of VDUP1 Inhibition Studies:

The study focuses on VDUP1 deficiency in UC; however, inhibiting VDUP1 activity could provide additional insights into its impact on inflammation and macrophage activation.

10)      Overemphasis on VDUP1 as a Therapeutic Target:

The study proposes VDUP1 as a therapeutic target for UC without addressing potential adverse effects or off-target effects of targeting VDUP1 in clinical settings. A more comprehensive evaluation of the therapeutic potential is needed.

11)      No Assessment of Causal Relationship:

The study identifies an association between VDUP1 deficiency and UC severity but does not establish a causal relationship between them, leaving room for alternative explanations.

12)      Absence of Comparison with Existing Literature: The discussion lacks a comprehensive comparison of the study's findings with existing literature on VDUP1 in other inflammatory conditions or in different disease models.

 12)  Acknowledgement of Limitations:

The discussion does not explicitly acknowledge the study's limitations, potentially leading to an overly optimistic interpretation of the results.

The quality of English used in the article is generally good, but there are some areas where improvement could be made. The sentences are mostly coherent and well-structured, and the scientific terminology is appropriately used. However, there are occasional grammatical errors and awkward phrasings that could be refined to enhance the overall clarity and fluency of the language.

The article could benefit from additional proofreading and editing to address these minor issues. Some sentences might be rephrased to improve readability, and careful attention to subject-verb agreement, prepositions, and verb tenses would further strengthen the quality of the English used.

While the language is understandable and conveys scientific information effectively, further polishing would enhance the article's presentation and ensure a more professional and polished tone. Overall, with some minor improvements, the quality of English in the article can be elevated to match the scientific content's importance and validity.

Author Response

Jong Soon Kang

Laboratory Animal Resource Center, Korea Research Institute of Bioscience and Biotechnology

30 Yeongudanji, Cheongwon, Cheongju 28116, Chungbuk, Korea

kanjon@kribb.re.kr

August 24, 2023

Editor’s name: Ms. Maeve Wang

International Journal of Molecular Science

Subject: Response to reviewer’s comments on Manuscript ID: ijms-2547718 [VDUP1 deficiency promotes the severity of DSS-induced Colitis in mice by inducing macrophage infiltration]

Dear Ms. Maeve Wang

I hope this e-mail finds you well.

I am writing in response to the reviewers' comments on our manuscript titled “VDUP1 deficiency promotes the severity of DSS-induced Colitis in mice by inducing macrophage infiltration”, which was submitted to IJMS under manuscript ID “ijms-2547718”. We are grateful for the time and effort the reviewers and the editorial team have dedicated to the evaluation of our work.

We would like to express our sincere appreciation for the constructive feedback provided by the reviewers. Their insights have undoubtedly enhanced the quality of our manuscript. We have carefully considered each comment and suggestion and have made the necessary revisions to address the concerns raised.

We believe that these revisions have strengthened the overall clarity, validity, and significance of our findings.

 Enclosed, please find our revised manuscript along with a detailed response to each of the reviewers’ comments. We have also highlighted the changes made in the revised manuscript for your convenience.

 We kindly request that you consider our revised manuscript for further review. We are confident that the revisions have substantially improved the manuscript, and we believe that our work aligns well with the scope and standards of IJMS. We are hopeful that the reviewers will find our responses and revisions satisfactory.

Thank you for considering our revised manuscript for reconsideration. We look forward to hearing from you and the reviewers regarding the outcome of this review process.

Please feel free to contact me if you require any further information or have any questions. Thank you for your time and consideration.

Sincerely,

Jong Soon Kang

Laboratory Animal Resource Center, Korea Research Institute of Bioscience and Biotechnology

30 Yeongudanji, Cheongwon, Cheongju 28116, Chungbuk, Korea

kanjon@kribb.re.kr

August 18, 2023

Response to Reviewer 1 Comments

Point 1. Small Sample Size and Lack of Replication: The study does not mention the sample size used in some experiments, which raises concerns about statistical power. Additionally, in some experiments, the number of replicates used is either low or not mentioned at all, which is making it difficult to assess the statistical robustness of the findings.

Response 1: We appreciate your comment on point 1. It seems like it will be beneficial in enhancing our article. We have checked and indicated the sample size and the number of replicates for all experiments in the revised manuscript.

Point 2. Lack of Detailed Experimental Protocols: The study lacks comprehensive descriptions of the experimental protocols, including details on sample preparation, RNA isolation, and qRT-PCR conditions. Insufficient information may hinder reproducibility and interpretation of results. In the gene expression analysis by QRT-PCR, B-actin was selected as the housekeeping (reference) gene for relative expression quantification. However, Nour Eissa et al 2017 recently reported that B-actin and Gapdh is less stable and highly variable housekeeping (reference) gene in DSS-induced colitis C57BL/6J mice, and TATA-box-binding protein (Tbp) and eukaryotic translation elongation factor 2 (Eef2) are most suitable as housekeeping (reference) genes. https://www.nature.com/articles/srep42427. The studies show that intestinal inflammation significantly affects the stability of mucosal Gapdh, Actb and β2m expression, which displays high variability in healthy individuals and/or between the non-inflamed and inflamed mucosa and that normalization of colonic TNF-α and IL-1β mRNA levels significantly dependents on the stability of reference genes. Therefore, the author should re-qualify the gene expression with a more appropriate reference gene or genes.

Response 2.

  • According to your opinion, we have supplemented the content for section 4.3.
  • We appreciate your comment on endogenous control. In this study, we employed beta-actin as the endogenous control for qRT-PCR. As you mentioned, Eissa et al showed the expression level of beta-actin in the colon of control animals and DSS-induced colitis and the p-value between control and DSS-induced colitis on beta-actin was 0.1282 which is not so much different from the p-value of Eef2 and means that the difference is not significant. Therefore, it is assumed that both Eef2 and beta-actin are proper endogenous controls in DSS-induced colitis model.

Point 3. Lack of Control Groups: The study lacks control groups for some analyses, which could limit the ability to draw robust conclusions. Including appropriate control, groups would strengthen the experimental design.

Response 3. We appreciate your comment on the lack of a control group. We have considered your opinion and revised experimental results throughout the manuscript.

Point 4. Insufficient Method Validation: The study does not provide validation of key methods used, such as the chemotaxis assay and TUNEL assay. Validation experiments would enhance the reliability of the results.

Response 4. We appreciate your feedback on method validation. We have validated experimental methods and previously reported several times. [TUNEL assay (Scientific Reports 13(1), Theranostics 2023; 13(8):2693-2709.) and chemotaxis assay (Int Immunopharmacol. 2018 Oct; 63:66-73., Eur J Immunol. 2006 May;36(5):1285-95.).]

 Point 5. Limited Mechanistic Insights: The study does not provide detailed mechanistic insights into how VDUP1 deficiency leads to increased inflammation, macrophage infiltration, and NF-κB activation. A more in-depth analysis of the underlying molecular mechanisms would strengthen the study's conclusions.

Response 5: We appreciate your feedback on mechanistic insights. Based on the suppression of VDUP1 expression in clinical samples of UC patients (Oncol Rep. 2007 Sep;18(3):531-5.), this study focused on investigating the preclinical features of ulcerative colitis in relation to VDUP1 deficiency, using experimental animals in the context of translational research perspective. We are going to consider your valuable opinion in our future studies.

Point 6. Limited Investigation of VDUP1 Activation: While the study focuses on VDUP1 deficiency, it fails to explore the role of VDUP1 activation. A comprehensive understanding of both VDUP1 deficiency and activation could provide a more holistic view of VDUP1's involvement in UC.

Response 6. We appreciate your feedback on VDUP1 activation. As we mentioned in response 5, this study originated from clinical features of VDUP1 suppression in UC patients. Therefore, we focused on VDUP1 deficiency-mediated preclinical features of ulcerative colitis in the DSS-induced colitis model. We are going to consider your valuable opinion in our future studies.

Point 7. Insufficient Discussion of Conflicting Findings: The study acknowledges the controversial role of VDUP1 in inflammation-related disorders but does not thoroughly discuss the possible reasons for these conflicting findings in different diseases.

Response 7. We appreciate your feedback on the discussion of contradictory findings. We added the information on the discussion of conflicting findings in the revised manuscript.

Point 8.  Incomplete NF-κB Activation Analysis: The study reports increased NF-κB p65 activation in VDUP1-KO mice, but a more in-depth analysis of the downstream targets and functional consequences of NF-κB activation would provide a clearer understanding of its role in UC pathogenesis.

Response 8. We appreciate your feedback on the incomplete NF-κB activation analysis. As we mentioned in response 5, this study focused on VDUP1 deficiency-mediated preclinical features of ulcerative colitis in the DSS-induced colitis model. We will consider your valuable opinion in our future studies using.

Point 9. Lack of VDUP1 Inhibition Studies: The study focuses on VDUP1 deficiency in UC; however, inhibiting VDUP1 activity could provide additional insights into its impact on inflammation and macrophage activation.

Response 9. We appreciate your feedback on the lack of VDUP1 inhibition studies. There are inhibitors that suppress the expression of VDUP1, but commercial inhibitors that selectively target VDUP1 activity are currently unavailable. After the VDUP1 inhibitor is commercially available, we will test the impact of the VDUP1 inhibitor on inflammation and macrophage activity.

Point 10. Overemphasis on VDUP1 as a Therapeutic Target: The study proposes VDUP1 as a therapeutic target for UC without addressing potential adverse effects or off-target effects of targeting VDUP1 in clinical settings. A more comprehensive evaluation of the therapeutic potential is needed.

Response 10. We appreciate your feedback on the overemphasis on VDUP1 as a therapeutic target. We have refined the wording in both the abstract and the discussion in the revised manuscript.

Point 11. No Assessment of Causal Relationship: The study identifies an association between VDUP1 deficiency and UC severity but does not establish a causal relationship between them, leaving room for alternative explanations

Response 11. We appreciate your feedback on the no assessment of the causal relationship. In this study, we provide evidence from a translational perspective that, similar to human UC samples, the expression of VDUP1 is reduced in the DSS-induced colitis model. Additionally, we present information indicating that the VDUP1 deficiency exacerbates UC. We will consider your valuable opinion in our future studies using DSS-induced colitis and UC-associated diseases.

Point 12. Absence of Comparison with Existing Literature: The discussion lacks a comprehensive comparison of the study's findings with existing literature on VDUP1 in other inflammatory conditions or in different disease models.

Response 12. We appreciate your feedback on the discussion of contradictory findings. We have incorporated the comparison with existing literature in the revised manuscript.

Point 13. Acknowledgment of Limitations: The discussion does not explicitly acknowledge the study's limitations, potentially leading to an overly optimistic interpretation of the results.

Response 13. We appreciate your feedback on the acknowledgment of limitations. We have incorporated the limitations of this study in the revised manuscript.

Reviewer 2 Report

Overall, this is an interesting paper that investigates the VDUP1 deficiency in the severity of colitis in mice. The authors evaluated the disease activity index, histological scores, number of globet cells, apoptotic index, mucin mRNA expression, proinflammatory cytokines expression, phosphorylated NF-kB expression, and infiltration of macrophages into the site of ulceration in Wild type and KO- VDUP1 mice in a DSS-colitis model. They found that DSS increased the severity of colitis in KO- VDUP1 mice more than the Wild type.

Overall, the concept of the studies is significant, and the manuscript is well written.

With the idea to help to improve the impact of the findings reported here I suggest taking into consideration the next,

  • The body weight data and DAI should be analyzed with a 3-way ANOVA instead of a 2-way ANOVA. This is because there are three variables that are changing: Wild type vs KO, healthy vs inflamed, and time (0-9 days). A 3-way ANOVA will allow the authors to test for interactions between these variables, which could provide more insights into the role of VDUP1 in colitis.
  • For figure 2A, the labels of the meaning of each group should be added, as shown in figure 2B. This will make the figure easier to understand.
  • The size of the lettering and the size of figure 4 and 5A should be increased. This will make the figures easier to read.

Author Response

Jong Soon Kang

Laboratory Animal Resource Center, Korea Research Institute of Bioscience and Biotechnology

30 Yeongudanji, Cheongwon, Cheongju 28116, Chungbuk, Korea

kanjon@kribb.re.kr

August 24, 2023

Editor’s name: Ms. Maeve Wang

International Journal of Molecular Science

Subject: Response to reviewer’s comments on Manuscript ID: ijms-2547718 [VDUP1 deficiency promotes the severity of DSS-induced Colitis in mice by inducing macrophage infiltration]

Dear Ms. Maeve Wang

I hope this e-mail finds you well.

I am writing in response to the reviewers' comments on our manuscript titled “VDUP1 deficiency promotes the severity of DSS-induced Colitis in mice by inducing macrophage infiltration”, which was submitted to IJMS under manuscript ID “ijms-2547718”. We are grateful for the time and effort the reviewers and the editorial team have dedicated to the evaluation of our work.

We would like to express our sincere appreciation for the constructive feedback provided by the reviewers. Their insights have undoubtedly enhanced the quality of our manuscript. We have carefully considered each comment and suggestion and have made the necessary revisions to address the concerns raised.

We believe that these revisions have strengthened the overall clarity, validity, and significance of our findings.

 Enclosed, please find our revised manuscript along with a detailed response to each of the reviewers’ comments. We have also highlighted the changes made in the revised manuscript for your convenience.

 We kindly request that you consider our revised manuscript for further review. We are confident that the revisions have substantially improved the manuscript, and we believe that our work aligns well with the scope and standards of IJMS. We are hopeful that the reviewers will find our responses and revisions satisfactory.

Thank you for considering our revised manuscript for reconsideration. We look forward to hearing from you and the reviewers regarding the outcome of this review process.

Please feel free to contact me if you require any further information or have any questions. Thank you for your time and consideration.

Sincerely,

Jong Soon Kang

Laboratory Animal Resource Center, Korea Research Institute of Bioscience and Biotechnology

30 Yeongudanji, Cheongwon, Cheongju 28116, Chungbuk, Korea

kanjon@kribb.re.kr

August 18, 2023

Response to Reviewer 2 Comments

Point 1. The body weight data and DAI should be analyzed with a 3-way ANOVA instead of a 2-way ANOVA. This is because there are three variables that are changing: Wild type vs KO, healthy vs inflamed, and time (0-9 days). A 3-way ANOVA will allow the authors to test for interactions between these variables, which could provide more insights into the role of VDUP1 in colitis.

Response 1. We appreciate your comment on point 1. It seems like it will be beneficial in enhancing our article. We have changed the statistical analysis on Figure A and supplement Figure S1 to a 3-way ANOVA instead of a 2-way ANOVA. We changed the wording on the legend for Figure A and “4.7. Statistical analysis” as well, according to the changing statistical analysis.

Point 2. For Figure 2A, the labels of the meaning of each group should be added, as shown in figure 2B. This will make the figure easier to understand.

Response 2. We appreciate your comment on point 2. We added the labels of the meaning of each group for Figure 2A.

Point 3. The size of the lettering and the size of figure 4 and 5A should be increased. This will make the figures easier to read.

Response 3. We appreciate your comment on point 3. We increased the size of the lettering and the size of Figure 4 and 5A.

Round 2

Reviewer 1 Report

The authors have made the suggested changes and answered my queries. I have no further questions/concerns.

1) Just a few points that need attention: The figures' quality is pretty low and needs some attention.

2) Figures 2a : Body weight (% of start ) should be replaced with % of Body weight changes.. Replace in 2a and 2b: Time (days) with just "Days". The legends in 2a need to be fixed.

3) Use colourblind-friendly text colour for figures 

Major concern:

The author must improve the results section of Figures 2a and 2b. Put the % of body weight changes and explain how much each group's body weight changed compared to the control group. Similarly, for gene expression studies instead of showing just the p-value, try to figure out the % changes and incorporate it in the result sections. It will significantly improve the manuscript and will attract the reader's attention. 

Author Response

Jong Soon Kang

Laboratory Animal Resource Center, Korea Research Institute of Bioscience and Biotechnology

30 Yeongudanji, Cheongwon, Cheongju 28116, Chungbuk, Korea

kanjon@kribb.re.kr

August 29, 2023

Editor’s name: Ms. Maeve Wang

International Journal of Molecular Science

Subject: Response to reviewer’s comments on Manuscript ID: ijms-2547718 [VDUP1 deficiency promotes the severity of DSS-induced Colitis in mice by inducing macrophage infiltration]

Dear Ms. Maeve Wang

I hope this e-mail finds you well.

I am writing in response to the reviewers' round 2 comments on our manuscript titled “VDUP1 deficiency promotes the severity of DSS-induced Colitis in mice by inducing macrophage infiltration”, which was submitted to IJMS under manuscript ID “ijms-2547718”. We are grateful again for the time and effort the reviewers and the editorial team have dedicated to the evaluation of our work.

We have finalized the content reflected in revision 1 and additionally indicated the changes made for revision 2 in the manuscript using blue text.

Thank you again for considering our revised manuscript for reconsideration. We look forward to hearing from you and the reviewers regarding the outcome of this review process.

Please feel free to contact me if you require any further information or have any questions. Thank you for your time and consideration.

Sincerely,

Jong Soon Kang

Laboratory Animal Resource Center, Korea Research Institute of Bioscience and Biotechnology

30 Yeongudanji, Cheongwon, Cheongju 28116, Chungbuk, Korea

kanjon@kribb.re.kr

August 29, 2023

Response to Reviewer 1 Comments

Point 1. Just a few points that need attention: The figures' quality is pretty low and needs some attention. Figures 2a : Body weight (% of start ) should be replaced with % of Body weight changes.. Replace in 2a and 2b: Time (days) with just "Days". The legends in 2a need to be fixed.

Response 1. We appreciate your comment on the figures’s quality being pretty low and the need for the replacement of legends in Figure 2A and 2B. It definitely enhanced our article. We have checked and changed the Figures according to your mention in the revised manuscript.

Point 2. Use colourblind-friendly text colour for figures 

Response 2. We highly appreciate your suggestion and have changed the captions of Figure 3A and Figure 5B to use a colorblind-friendly text color in the revised manuscript.

Point 3. Major concern: The author must improve the results section of Figures 2a and 2b. Put the % of body weight changes and explain how much each group's body weight changed compared to the control group. Similarly, for gene expression studies instead of showing just the p-value, try to figure out the % changes and incorporate it in the result sections. It will significantly improve the manuscript and will attract the reader's attention. 

Response 3. We appreciate your comment on adding the percentage of body weight changes and explaining how much each group’s body weight changed compared to the control group. We have considered your opinion on “including % of body weight changes and explain how much each group’s body weight changed compared to the control group”. We have also adapted in that manner to the mRNA expression and revised experimental results throughout the manuscript.  
